# A Brief Review of Photocatalytic Reactors Used for Persistent Pesticides Degradation

**Gabriela Olimpia Isopencu** [ID], **Alexandra Mocanu** [ID] **and Iuliana-Mihaela Deleanu** *[ID]

Department of Chemical and Biochemical Engineering, University Politehnica of Bucharest, Polizu Str. 1-7, 011061 Bucharest, Romania
* Correspondence: iuliana.deleanu@upb.ro

**Abstract:** Pesticide pollution is a major issue, given their intensive use in the 20th century, which led to their accumulation in the environment. At the international level, strict regulations are imposed on the use of pesticides, simultaneously with the increasing interest of researchers from all over the world to find methods of neutralizing them. Photocatalytic degradation is an intensively studied method to be applied for the degradation of pesticides, especially through the use of solar energy. The mechanisms of photocatalysis are studied and implemented in pilot and semi-pilot installations on experimental platforms, in order to be able to make this method more efficient and to identify the equipment that can achieve the photodegradation of pesticides with the highest possible yields. This paper proposes a brief review of the impact of pesticides on the environment and some techniques for their degradation, with the main emphasis on different photoreactor configurations, using slurry or immobilized photocatalysts. This review highlights the efforts of researchers to harmonize the main elements of photocatalysis: choice of the photocatalyst, and the way of photocatalyst integration within photoreaction configuration, in order to make the transfer of momentum, mass, and energy as efficient as possible for optimal excitation of the photocatalyst.

**Keywords:** photoreactor; persistent pesticide; photocatalytic degradation; photocatalysis mechanism; slurry photocatalyst; immobilized photocatalyst

## 1. Introduction

A Plant Protection Product (PPP) known as a pesticide is a chemical compound or a mixture of different active ingredients that act as a "fortification" against pests and plant diseases [1,2]. Regardless of their classification [3], pesticides are most commonly used in agricultural and health sectors [4]. Herbicides, fungicides, insecticides, or rodenticides which are the commercial terms for different pesticides could have a great action against insects, plant pathogens, or weeds thus preventing crop yield losses or enhancing crop productivity [5–7]. Despite their crucial role in the agricultural industry and farming, pesticides may cause dramatic effects to the environment and human health [8–11].

Humans can get exposed to pesticides by respiratory, eye, dermal, or oral pathways. Traveling through the bloodstream in the whole human body, the pesticides can affect different organs causing reversible or irreversible effects [10,12]. Although pesticides can be eliminated from the human body by urine, respiration (by exhaling), or skin, the impact on human health depends on the exposure time, the concentration of the pesticide, and the sensitivity of certain persons to these compounds [10]. Thus, asthma, diabetes, Parkinson's disease, leukemia, and cancer were more often registered in the case of professional pesticide applicators or agricultural workers compared with the population that was exposed to pesticides through the food chain, contaminated water, air, or soil [10,13–15].

Based on the health risk and toxic action of pesticides for the population, the World Health Organization (WHO) estimated the $LD_{50}$ (median lethal dose) to include pesticides into four hazardous categories. Thus, the pesticides were classified as extremely toxic,

highly toxic, moderately toxic, and slightly toxic by exposing rats to oral and dermal contamination [16,17].

In European Union, in order to diminish the diseases associated with pesticides exposure, several authorities like European Commission (EC), European Chemical Agency (ECHA), and European Food Safety Authority (EFSA) have the legislative role to control and monitor the approval and use of pesticides for industrial and householding activities. Thus, a PPP is principally regulated by Regulation (EC) No. 1107/2009 which provides information on how pesticides can be used according to their role in the cultivated field thus helping the development of crops by killing other competing plants or harmful organisms [18]. Due to the harmful action of pesticides on long-term exposure discussed before the maximum residue levels (MRL) for pesticides were established for food by Regulation (EC) No. 396/2005, Article 32, while the sustainable use of pesticides is controlled by the Directive 2009/128/EC [19]. Annually, the European Union publishes a report on pesticides residues in different food products carried out by EU Member States, Iceland and Norway revealing not only the limit of residues, but also detecting pesticides that were not approved by EU legislation so far providing information about restrictive use of different classes of pesticides [20].

Although these regulations and rules are considerably useful in terms of pesticide use, in some cases, it is still not clear to what extent pesticides can induce genetic changes not only in humans, but also in other living organisms like invertebrates, fishes, reptiles, birds, mammals, or bees currently due to the contamination of their environment with toxic pesticides [21–25]. Computer models or statistics failed to correlate the hazardous effect of pesticides at a laboratory scale with those in the field in which ecological factors and metabolic processes contribute to the enhancement of the toxicological effects of different pesticides [3,26].

In 2017, in the EU, 78 pesticides were authorized for crop treatment. Since then, other pesticides were banned and others are in the process of being banned, and farmers are facing serious problems with certain agricultural activities since the number of safe considered pesticides is constantly reduced [27].

Among the top 20 crop-specific pesticides that were mostly used since 2015, glyphosate and atrazine are herbicides with a large spectrum against weeds. Analysis of contaminated drinking water with atrazine concluded that this herbicide could alter the human hormonal system and it was classified as an endocrine disruptor [28,29]. As a consequence, it was banned from the EU in 2004 [29], while the status of glyphosate is still controversial and will face a new evaluation at the end of 2022. Currently, glyphosate is the most widely used herbicide in the EU due to its efficiency against annual and perennial weeds.

Its controversial status resides from the reports of the International Agency for Research on Cancer (IARCs) in 2015 that concluded that glyphosate is probably carcinogenic to humans although this compound was on the list of approved pesticides since 2002 in the EU and extended approvals were taken into consideration by the European Commission in the last years.

The major issue was registered in 2015 for some products in which glyphosate was the active ingredient, but polyethyloxylated tallow amine surfactant was used in the formulation, classified meanwhile as carcinogenic and banned, thus concluding that glyphosate could not be classified as carcinogenic. However, actions developed by various NGOs imposed a new approval for December 2022, although, in the USA, Australia, Canada, and New Zealand glyphosate was not included on the list of carcinogenic pesticides [27]. Currently, the glyphosate issue is crucial since this herbicide represents more than 25% or 33% of the total herbicide use in countries like France, the United Kingdom, Sweden, and Germany and a possible ban not only will force farmers to change their strategy in crop cultivation and rotation but will generate an income reduction up to 8% in Sweden and almost 14% in the United Kingdom [27,30].

Nevertheless, the possibility of pesticide residues contaminating potable water, soil, and air, and produce harmful effects on the environment still remains a great challenge.

Thus, in the last years, many researchers tried to give different solutions in terms of water, soil, or air decontamination. From this point of view, the focus of this review is targeted mostly on the most used pesticide decontamination possibilities by photocatalytic processes.

Photocatalytic processes are green advanced techniques consisting of the usage of light to modify the reaction rate of a process, in the presence of a photocatalyst [31]. By applying these kinds of processes, two major directions can be pursued: (i) Organics can be oxidized aiming synthesis, conversions, or complete degradation/mineralization (in water and wastewater treatment); (ii) solar energy can be transformed into sustainable fuels, like hydrogen through water splitting [32,33].

This paper will take into consideration the catalyzers' efficiency, the general mechanism of pesticide degradation, and especially different configurations of the plant units, under various experimental conditions. It is well known that designing an optimal photocatalytic process/system depends on a wide spectrum of independent and interdependent variables: pH, catalyst dose, irradiation intensity and type, pollutants concentration and type, reaction time, temperature, mixing conditions, and so on [34,35]. The purpose of this review is to bring into light, in a novel approach, the current progress in terms of pilot or industrial scale-up of the pesticide photocatalytic decontamination processes, using different reactors, designed not only for single pesticide solutions, but also for pesticide mixtures. Based on the diversity of configurations and versatility of such systems, another aim of this review is to identify future perspectives related to the efficiency of slurry versus immobilized reactors for water remediation.

## 2. Mechanisms of Pesticides Photochemical Degradation

### 2.1. General Considerations

A photochemical reaction is a process that must be preceded by the absorption of radiation of the appropriate energy by a molecule. Upon absorption of radiation, the excited molecule can be transformed, in one or more steps, into a product or it can be transformed into an intermediate species that can participate in subsequent reactions of a thermal nature, as happens, for example, in the chain reactions. Sometimes absorption of radiation occurs in one molecule, but definitive changes occur in others, as in photosensitized and photocatalyzed reactions.

Photochemical reactions are attractive because photoactivation can be highly selective. This advantage is the result of a precise and controlled change in the electronic state of a molecule through the absorption of radiation, due to the fact that photocatalytic redox reactions take place on the surface of a semiconductor exposed to UV/visible radiation.

This selectivity of the catalytic process is also combined with general conditions of the process because the excitation of the reactant is achieved by radiation energy with very weak heating abilities, and consequently, the photochemical reactions do not involve high temperatures, nor are they generally required due to the activation mechanism involved.

The stages of the photocatalytic general mechanism are the adsorption of reactants on the surface of the catalyst, catalytic action on reactants, and desorption of obtained products [36]. Photocatalysis mechanisms suppose the incidence of a photon of light having energy comparable to the band gap energy of a semiconductor on its surface, and the result is an electron-hole pair.

This mechanism based on photo-induced electrons and holes determines the reaction with oxygen ($O_2$), water ($H_2O$), and hydroxyl groups to generate reactive oxygen species (ROS) such as hydroxyl radicals ($\cdot OH$) and radical anions of superoxide ($\cdot O_2^-$) with strong oxidation abilities. These ROS are the main species responsible for the degradation of persistent organic pollutants in wastewater. These charge carrier species help to degrade toxic chemical species. Both organic and inorganic pollutants get degraded with this green technology. The key advantage of this method is that no special oxidant is required for the reaction, as atmospheric oxygen itself acts as a good oxidant. [36,37].

In the process of agro-wastewater reclamation, the mineralization of the pesticides is the goal. Here also, the central oxidation specie is non-selective (·OH), but the less reactive free radical ($H_2O$·) and its conjugate ($O_2$·) have their contribution as well. The pesticides react with the free radicals by hydrogen abstraction or electrophilic addition to double bonds. The radicals further react with $O_2$, resulting in ($ROO^-$), organic peroxyl radicals. Numerous distinctive intermediates are forming until total mineralization, achieved through different oxidation paths [38].

However, there are some shortcomings of these photocatalysts, such as wide band energies (Eg), low light absorption abilities, and fast recombination rates of photon-induced electrons and holes, which have limited the use of this catalytic process [39].

The photocatalytic oxidation of pollutants requires high potential, thus the valence band location at the semiconductor-electrolyte interface must be more positive, as exhibited by $TiO_2$ or CdS for the photogenerated holes to have sufficient energy to oxidize the organic pollutants through the generation of hydroxyl radicals. The redox potential must lie within the band gap of the photocatalyst [40,41].

As presented in Figure 1, the possible structures of heterojunctions are classified into three categories based on their band gap energies [40,42]:

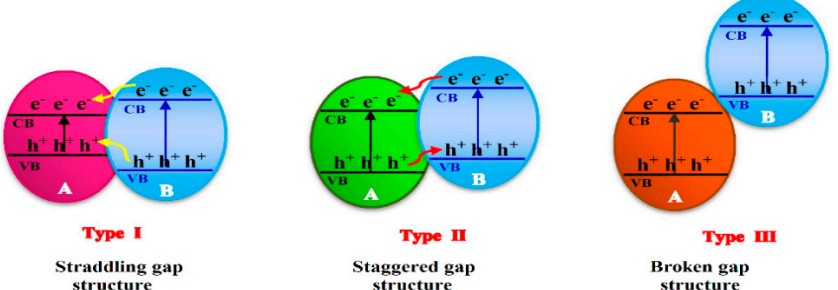

**Figure 1.** The three types of heterojunction structure of photocatalyst semiconductors [40,42].

I. In straddling gap structure (type 1), the conduction band (CB) of semiconductor B is more negative than semiconductor A, and its valence band (VB) is more positive than semiconductor A. Based on the principle of charge carrier transfer, electrons and holes will accumulate in smaller Eg in semiconductor A. Consequently, charge carrier recombination might occur due to smaller Eg, which will reduce photocatalytic performance.

II. In a staggered gap structure (type 2), the CB of semiconductor B is more negative than semiconductor A, and the VB of semiconductor A is more positive than semiconductor B. Therefore, electrons will transfer from semiconductor B to semiconductor A while holes transfer from semiconductor A to semiconductor B. The electron-hole separation for type 2 is better than type 1 because charge carriers are separated into two semiconductors.

III. In a broken gap structure (type 3), both CB and VB of semiconductor A are lower than semiconductor B. Therefore, both electrons and holes are not able to pass the interface to the respective bands in the semiconductor. This is because the transport of charge carriers at the interface is interrupted by the energy barrier.

Among the three heterojunction structures, the type 2 semiconductor heterojunction structure is the most typical heterojunction system [40,42].

Despite their advantages, photochemical reactions are not widely used in industrial practice. They have been adopted if no alternative thermal or catalytic process is available or the production scale is small and very often dedicated to high-added value products; then, processing difficulties and the negative effects of operating and equipment costs are greatly reduced.

Sustained studies have been carried out to control and stabilize the photocatalytic effect more efficiently by developing new types of photocatalysts, including hybrid materials, with the aim of modifying the kinetics of electron transfer in order to obtain a large dipole moment and more electrons to be transferred from the VB to CB of the photocatalyst,

leading to a narrowing of the Eg band value which may lead to better absorbance in visible light or natural sunlight. However, these two goals are conflicting and are thus difficult to realize simultaneously in a single-component photocatalyst [40]. In order to improve some of these drawbacks, the synthesis process was directed to the development of photocatalysts with structural controlled defects or surface defects [43]. The structural imperfections are responsible for the extension of the light absorption wavelength range, while the surface defects act like active sites for the catalytic reactions [44].

In our opinion, some efforts that were carried out to elucidate part of the photocatalytic mechanism led to this new generation of photocatalytic systems with controlled defects that enhanced the photodegradation process of the polluting compounds in general, and pesticides in particular.

### 2.2. Specific Mechanisms

There are, as emphasized before, numerous factors that influence photocatalytic degradation. Furthermore, the process is strongly affected by a large number of ions already presented in water/wastewater ($Cl^-$, $SO_4^{2-}$, $NO_3^-$, $Fe^{3+}$), so for each organic component/pollutant, different degradation pathways have been found, depending on operational conditions. In most cases, a complex degradation mechanism is involved, which needs further investigations [45].

Just to give some insights on the process complexity, we use as an example glyphosate, a most extensively used herbicide in the world, once considered environmentally friendly. Numerous degradation pathways were investigated and reported for glyphosate-based herbicides. In a comprehensive summary, Feng et al. presented its potential oxidation pathways under different processes [46]. Figure 2, presented here, shows that glyphosate photo-degradation is in most cases related to aminomethylphosphonic acid (AMPA) and sarcosine pathways.

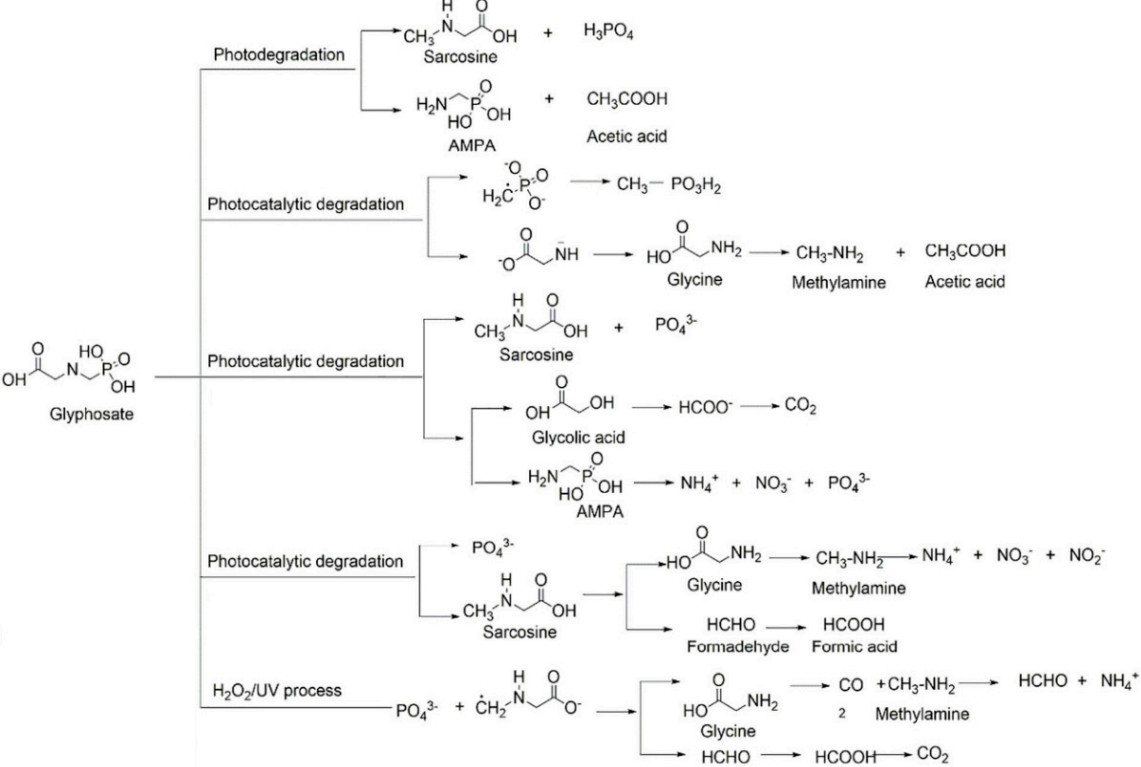

**Figure 2.** Glyphosate photodegradation pathways (adapted after [46]).

Similar, complex degradation pathways can be found in literature for atrazine [47], or other pesticides [48].

## 3. Photocatalyst Used for Pesticides Photodegradation

Photocatalysts are substances responsible for harnessing solar energy for the degradation of persistent organic pollutants by absorbing light in water. Photocatalytic substances have appeared since 1972 discovered by Fujishima and Honda when they highlighted the photocatalytic effect of titanium dioxide ($TiO_2$), which catalyzes the splitting of water into hydrogen and oxygen in a photo-electrochemical cell [40]. Thus, photocatalysts are supported or un-supported semiconductors that use light to catalyze chemical reactions. A photocatalyst's characteristics must be photoactivity, photostability, and capability of utilizing UV/visible light, to be biologically and chemically inert, nontoxic, and accessible [36,49].

### 3.1. Pure and Mixed Oxide Semiconductors

Numerous semiconductors, such as $Fe_2O_3$, $SnO_2$, $SrTiO_3$, $TiO_2$, $TiWO_5$, $WO_3$, $ZnO$, $WSe_2$, $CdS$, $Si$, etc., have been developed and used as photocatalysts. The interest in such applications started in 1972 when $TiO_2$ electrodes were used for water splitting [45].

Photocatalysts generally, can be synthesized from different categories of elements: noble metals, transition metals and non-metals, and metalloids depending on their physical and chemical properties. Examples of noble or rare metals are platinum (Pt), gold (Au), silver (Ag), palladium (Pd), ruthenium (Ru), cesium (Ce), rhodium (Rh), tungsten (W), and others. Transition metals include titanium (Ti), zinc (Zn) and copper (Cu), tin (Sn), strontium (Sr), while nitrogen (N), clay, graphene, and carbon dots (CD) are classified as non-metals and metalloids [39,50–53]. Among all the photocatalysts, $TiO_2$ and $ZnO$ are the most used bulk photocatalysts for the industrial photocatalysis processes, but currently, materials and matrixes for supporting the photocatalysts are being developed for maximum conversion efficiency of the polluting substrate, while conventional materials used for photocatalysis are dopped with metal ions to improve the photocatalytic efficacy [54–57].

Titanium dioxide is one of the most studied substances from the point of view of the photocatalytic effect since the discovery of its catalytic effect. $TiO_2$ exists in three different polymorphic forms: anatase, rutile, and brookite, with band gaps of 3.2, 3.0, and ~3.2 eV, respectively, activated in the UV range. Anatase and rutile are the most common polymorphs, with the anatase phase possessing higher photocatalytic activity than rutile and brookite [56,58,59].

### 3.2. Doped Photocatalysts

Another dominant trend in photocatalysts synthesis is related to doping procedures of semiconductors like $TiO_2$ or $ZnO$ with metallic and non-metallic elements (i.e., Cu, Fe, Sn, N, S, Ag, Au, etc.) to extend the wavelength absorption range in order to activate the photocatalytic process by solar light to degrade organic pollutants [60–62].

Different synthesis techniques were employed to obtain such materials like (a) Physical mixtures of preformed particles coming from the semiconductor material and the doping material; (b) Reduction of the doping agent directly on the surface of the semiconductor; (c) Impregnation of support with different salt precursors followed by evaporation of the solvent and calcination [61,63,64].

In our opinion, regardless of the synthesis method, the crucial role in increasing photocatalytic activity was related to the concentration of the doping agent, the morphology, and the size of the final photocatalysts.

Not only that the small concentration of the doping agent (i.e., 1–5% wt.) is enough to decrease the energy band of the semiconductors to limit the recombination of the electron-hole pairs, but also the dramatical changes in the crystallite size or shape of the final material led to enhanced photocatalytic activities even in visible light [32,65–68].

Based on these facts that brought to light the importance of the size of the photocatalysts particles, great attention should be given to photocatalytic nanomaterials.

*3.3. Nanomaterials*

Nanomaterials can also act as catalysts and produce better results than other structural types of photocatalysts. Nano-sized semiconductor photocatalysts gained great potential for removing large organic molecules such as dyes and pesticides in an environmentally friendly and sustainable manner [69–71]. Thus, nano-sized photochemical catalysts (i.e., CdS, ZnO, $TiO_2$, etc.) are embedded in new, environmentally friendly matrices such as bacterial cellulose, clay, complex organic structures, or graphene-based composites [55,72,73].

Another method to satisfy both the above-mentioned requirements, namely, reducing the band gap of the semiconductors while making the CB potential more negative and the VB potential more positive is Z-scheme photocatalytic systems. The Z-scheme photocatalysts are named as such because their charge transfer mechanism is similar to natural photosynthesis in green plants, in which the charge-carrier transport pathway involves a two-step photoexcitation that resembles the English letter "Z" (Figure 3) [41,74].

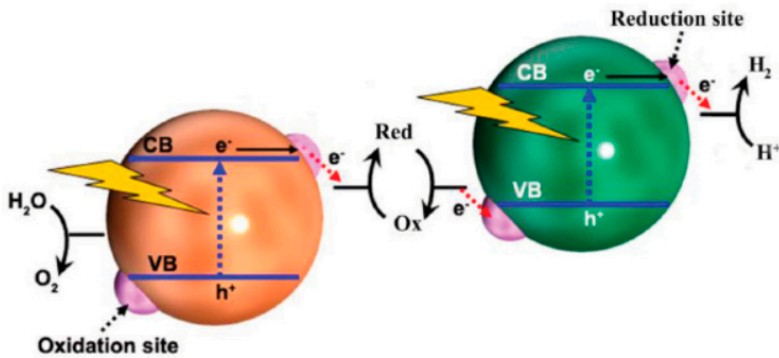

**Figure 3.** Schematic illustration of the Z-scheme [74].

Some examples of effective catalysts in the degradation of the most used pesticides will be presented further in our work.

Given their importance on photocatalytic process performances, the purpose of the present paper is to give insights on the recent status and future strategies on reactor configurations and operational parameters, emphasizing scale-up possibilities.

## 4. Photoreactors Types and Configurations for Pesticides Degradation–Design and Scale-Up Perspectives

Any photocatalytic setup, disregarding the scale, consists of reagents, a light source, and the reactor with its operating system [75]. New chemical synthesis routes, new catalysts, doping agents, and immobilization materials are continuously investigated and tested, but as summarized by Constantino et al. (2022), innovative apparatus/methods should provide solutions for process scale-up and efficiency [76].

As with any other conventional reactor, the photoreactor can be operated in batch or continuous regime, and this will be an important parameter in sizing the system. The similarities, however, end here since the fabrication of an effective photocatalytic reactor must consider several specific design parameters such as reactor geometry/area, radiation source, type of photo-catalyst, and operational parameters, among which (but not only): catalyst concentration, rate transfer of pollutants/reaction kinetics, light wavelength, and intensity, pH, temperature, dissolved oxygen concentration [77–79].

As complex as may be, there are some general aspects that can be emphasized: (i) Most used photocatalytic reactors, as reported in the literature, are batch or semi-batch-operated, characterized by small volumes (up to 100 L for some pilot-scale plants); (ii) The radiation source can be either natural (the sun), or artificial, consisting in UV-lamps; both are widely investigated and used even in the same system, but the second option will provide a constant radiation flux with the cost of additional energy consumption; (iii) Appropriate oxidations rates are obtained when the dissolved oxygen concentration is maintained near

to saturation. The oxygen can be supplied by direct contact with atmospheric air (for small reaction volumes) or by air compressors [80].

In a simplified approach, photocatalytic reactors can be divided into two major categories by the state of the photocatalyst, i.e., dispersed systems (stirred or stagnant slurry reactor) and immobilized systems. Although most of the reported studies indicate dispersed systems as more feasible and efficient, the immobilized systems are also investigated for the remediation of wastewater containing pesticides as will be further detailed.

### 4.1. Slurry Reactors for Pesticides Degradation

Also named suspended liquid reactors, slurry reactors can be successfully used for pesticide treatment and agro-wastewater reclamation.

The slurry photo-reactors contain the catalyst in the form of fine particles or nanoparticles suspended in the aqueous environment. The catalyst dispersion is determined by forced convection (provided by stirring) or by natural convection (due to an existing heat gradient for instance) [81]. The most important advantage to be mentioned, compared to other systems, is that slurry reactors offer a larger surface area of the catalyst. In many cases, because of very small particles, this advantage is doubled by a major drawback: difficult and expensive catalyst separation [82].

In search for less expensive configurations characterized by high optical efficiency and high quantum efficiency, continuous increasing attention has been given to solar reactors. The design of such systems should allow turbulent working regimes and should use direct and diffuse solar radiation at the same time. Compound Parabolic Concentrators (CPCs), initially used for solar concentration with static devices, are low-concentration collectors successfully implemented in the photochemical degradation of pesticides [83]. In an earlier report, two main pilot plants located in "Plataforma Solar de Almeria (PSA)" were indicated as large-scale facilities used to analyze solar light photodegradation potential for water detoxification [84]. Over the years pesticide destruction was investigated here. Thus, in 1993, Minero et al. [85] reported complete photocatalytic degradation of pentachlorophenol using $TiO_2$ slurry in small cylindrical glass cells under simulated solar light (1500 W Xenon lamp). The best conditions for process scale-up were assessed and applied effectively for the pesticide degradation in the large solar plant at PSA. The study is presenting a comprehensive description of the large-scale plant, operated close to the ideal plug flow reactor for more than 800 L suspension capacity. CPC modules were selected for pesticide degradations in further studies, operated as a perfectly agitated slurry batch reactor, with a reaction volume of up to 250 L. $TiO_2$ was generally used as a catalyst [86–88].

In 2006, Pérez et al. [89] studied at PSA comparative degradation of three pesticides, in four different photocatalytic approaches: heterogeneous with $TiO_2$, heterogeneous with $S_2O_8^{2-}$, photo-Fenton with $Fe^{2+}$, and photo-Fenton with $Fe^{3+}$, under similar experimental conditions: under sunlight, water flow 20 L/min, total volume 35 L (of which 22 L is total irradiated volume), batch operation. Complete mineralization was attained in all cases, but photo-Fenton using iron proved to be faster. The optimal solution for each case can be determined only by economic reasons, as technically any of the studied systems worked.

More recently, but using the same experimental setup, as presented here in Figure 4, Luna-Sanguino et al., reported the usage of hybrid photocatalysts for the photodegradation of a complex pesticide mix [90]. Hydrogen peroxide was tested as an additional oxidant agent and compared with oxygen from the air. Two $TiO_2$-rGO, titania-reduced graphene oxide nanocomposites were prepared using P25 Aeroxide® (P25) and Hombikat UV100 (HBK) and tested. Methomyl, pyrimethanil, isoproturon, and alachlor mix of pesticides was photodegraded. The results showed no major benefits of using $H_2O_2$ with any of the studied catalysts. HBK-rGO presented improved performances at low concentrations of pesticides (200 μg/L), achieving complete removal in less than 25 min [91].

Table 1 presents a comprehensive overview of reactor types and catalysts for the degradation of two of the most used pesticide active ingredients worldwide, according to PEST-CHEMGRIDS [92]. As can be seen, almost all of them are slurry reactors.

**Table 1.** Reactors type and photocatalytic conditions for pesticides degradation.

| Pesticide | Reactor Type | Catalyst | Light Source | Degradation | Ref. |
|---|---|---|---|---|---|
| Atrazine | Suspension, magnetically stirred | N,F-codoped $TiO_2$ NWs | $\lambda$ 365, 2.5 mW cm$^{-2}$, 15 W visible light irradiation, 15 W fluorescent lamps UV light irradiation, two 15 W UV light lamps (365 nm wavelength) | Not reported | [93] |
| | Suspension | N-$TiO_2$/ZSP | UVA-365 nm radiation was simulated by two 15 W lamps | | [94] |
| | Suspension, air bubbled | Au/$TiO_2$, Cu/$TiO_2$ and Ni/$TiO_2$ | UV–vis UV–PC lamp with primary emission at 254 nm | 60% | [95] |
| | Suspension, mechanically stirred | Fe$^{+3}$-$TiO_2$ | UV lamp protected by a Quartz tube | 99% | [96] |
| | Suspension | B-doped $TiO_2$ (A/R) | 350 W (15 A) Xenon lamp with a 300 nm cutoff filter | 94% | [97] |
| | Suspension | $[\alpha\text{-SiW}_{12}O_{40}]^{4-}$ $[\alpha\text{-PW}_{12}O_{40}]^{4-}$ $[P_8W_{48}O_{184}]^{40-}$ | Two 8 W UV-Xenon lamps, 254 and 366 nm | 56% 31% 41% | [98] |
| Glyphosate | Vertical annular photocatalytic reactor, air bubbled | $TiO_2$– $SiO_2$ monolithic aerogel | 16 W UV lamp (254 nm) | >99% | [99] |
| | Cylindrical batch reactor, suspension mixed by a peristaltic pump | W-Doped ZnO | Solar simulated lamps, 300–700 nm | 74% | [100] |
| | Continuous packed bed reactor | $TiO_2$ Degussa P25 $TiO_2$-Mn | UV lamp, 70 W, 370–410 nm | 28% 39% | [101] |
| | Plug flow reactor, suspension, magnetic stirrer | $TiO_2$ | 500 W high-pressure mercury lamp with mean wavelength 365 nm | 90% | [102] |
| | Suspension, magnetic stirring | Mn-doped-$TiO_2$ | Visible-light halogen linear lamp (500 W, 9500 Lumens) | 80% | [103] |
| | Cylindrical, suspension, mechanically stirred | $Zn_3V_2O_8$/40 wt% g-$C_3N_4$ | 300 W Xe lamp with a 400 nm cut-off filter Visible light intensity 180 mW cm$^{-2}$ | 85% | [104] |
| | Suspension, stirred and bubbled with oxygen | Ce–$TiO_2$ nanotubes | 125 W high-pressure mercury lamp | 76% | [105] |
| | Suspension, stirred | Goethite magnetite | Mercury UV lamp (CEL-M500/350, incident light intensity 500–2000 W/m$^2$, equipped with an optical filter for 275 nm) or a xenon Vis lamp (CEL-S500/350, incident light intensity 500–2000 W/m$^2$, wavelength 350–1100 nm) | 41% 71% | [106] |
| | Suspension, ultrasonic stirring | $Bi_2S_3$/$BiVO_4$(040) | Visible light irradiation ($\lambda$ > 400 nm), using a 125-W high-pressure mercury lamp with 180 mL of 2 mol/L $NaNO_2$ solution as the filter liquor | 79% | [107] |
| | Suspension, magnetic stirring | CDs/$MoS_2$/g-$C_3N_4$ | Simulated sunlight irradiation with AM 1.5 cut-off filters and the light intensity 1000 mw | 79% | [108] |

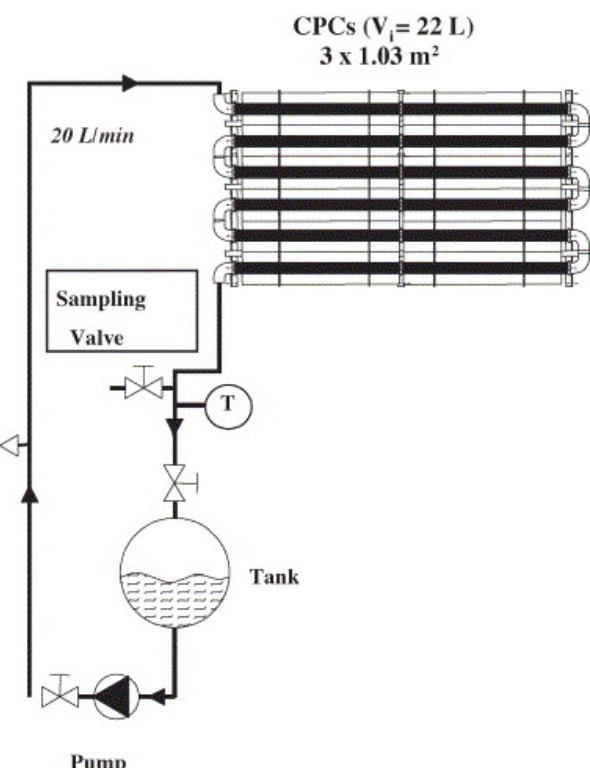

**Figure 4.** Schematic representation of the photoreactor with CPC modules [89].

Until now, the study and applications of slurry reactors were mainly based on experimental approaches, and very little based on mass transfer considerations or kinetic analysis. Mass transfer limitations can be determined by radiation gradients, concentration gradients in the bulk of the reaction volume, and by catalytic particles (or agglomerates) internal and external transport limitations. Ballari et al. reported investigations on this issue [109,110]. They found that non-uniformity of the irradiation area could result in significant concentration gradients that will produce mass transfer limitations in the bulk. Only very good mixing/fully turbulent flow conditions could create a proper environment for perfect mixing assumptions (meaning no concentration gradients) [82,109]. Also, only large particle sizes could determine external mass transfer limitations, while internal mass transfer in the solid phase (particles or agglomerates), determined by light penetration restrictions, will most likely occur [110]. A complete/mechanistic mathematical model to engineer an industrial photoreactor comprises of momentum and mass balance equations together with radiative transfer equations [111]. Due to photocatalysis specifics, which make "ideal" conditions impossible to attain, the kinetic analysis is particularly difficult: the radiation field is not uniform, and protons cannot be "mixed" as any other reactants [112]. Furthermore, intrinsic properties of the catalyst are essential to reactors design and optimization [113].

In our opinion, there are three major issues related to slurry systems: (i) The non-uniform irradiation of the photocatalyzer; (ii) The reduced capacity of the slurry systems to ensure the treatment of contaminated samples with high concentrations of pesticides; (iii) The difficulty to recover and reuse the photocatalytic system. Thus, the slurry systems need to be improved in terms of photoreactor configuration due to the limitation of UV or natural light to irradiate uniformly the whole photocatalytic active area. Also, creating photocatalysts more susceptible to natural/solar light could give a great advantage in terms of costs considering the possibility of removing UV or artificial light sources that could be replaced by solar light.

Nevertheless, the great advantages of this system remain the versatility of the slurry photoreactors, since they can be used for any type of organic pollutants, and the possibility

to scale-up easily compared to other types of photoreactors (for instance, those in which immobilization of the photocatalytic system is applied).

### 4.2. Immobilized Systems for Pesticides Treatment and Agro-Wastewater Reclamation

As an alternative to some of the drawbacks of slurry photoreactors, immobilized photoreactors have gained popularity due to the possibility of recycling the photocatalytic system. Thus, this section is dedicated to the immobilized systems for pesticide treatment and agro-wastewater reclamation.

Immobilized photoreactor refers to systems where the catalyst particles are grafted to proper substrates by impregnation, atomic layer deposition, spraying, or other methods as an effective approach for the catalyst's recovery and reusability, especially when nanoparticles are used [114]. Other important advantages, not yet fully recognized, are easier light penetration in the absence of high turbidity characterizing slurry systems [115], and simplicity for continuous operation (if applicable) [113].

Literature survey provides only a few studies of pesticide degradation in immobilized systems, and some are presented here.

Thus, Gar Alalm et al. studied the degradation of 2,4-dichlorophenol using a S-TiO$_2$ catalyst immobilized on a circular aluminum plate by polysiloxane, placed on the bottom of the reaction beaker [116]. The light source was a 400 W metal halide lamp. Degradation rates were obtained with the selected catalyst in both slurry and immobilized approaches. When immobilized, the catalyst required longer time to achieve comparable degradation rates with respect to the slurry, but it proved reusability and stability in multiple (five) sequential cycles.

Following previous limited studies on innovative floatable photo catalysts usage, Sivagami et al. investigated the degradation of three pesticides (Monocrotophos, Endosulfan, and Chlorpyriphos) using TiO$_2$ deposited on polymeric beads, under solar irradiation [117]. Relatively high removal rates were obtained in batch Immobilized Bead Photo Reactor, after 60 h of illumination, under magnetic stirring, performances being directly influenced by initial pesticide concentration and aeration degree.

Imidacloprid pesticide was degraded under UV light using nano-TiO$_2$ immobilized on a glass plate by heat attachment method [118]. The reactor was a borosilicate dish with a working volume of 400 mL, aq. solution. Processing parameters influence was investigated, and good removal rates, up to 90 %, were obtained for selected conditions.

An interesting design of immobilized systems was recently proposed and tested for real wastewater collected from an agrochemical source by Samy et al., 2021 [119]. In this innovative system, very high percentages of removal were obtained for diazinon in 60 min. Precisely, this novel photoreactor comprises two basins, each equipped with a stirrer, connected through an inclined surface, as can be seen in Figure 5.

Jiménez-Tototzintle et al. treated real agricultural wastewater in an Immobilized Biomass Reactor and applied photocatalysis as tertiary treatment for the bioreactor effluent. The photoreactor system comprises a continuously stirred tank, a recirculation pump (working flow 2.5 L/min), and a CPC module [120]. TiO$_2$ supported on glass beads was packed in two borosilicate glass tubes and was used as a catalyst under solar irradiation. The results indicate complete removal of thiabendazole and imazalil and almost complete removal of acetamiprid (92%) when hydrogen peroxide was added.

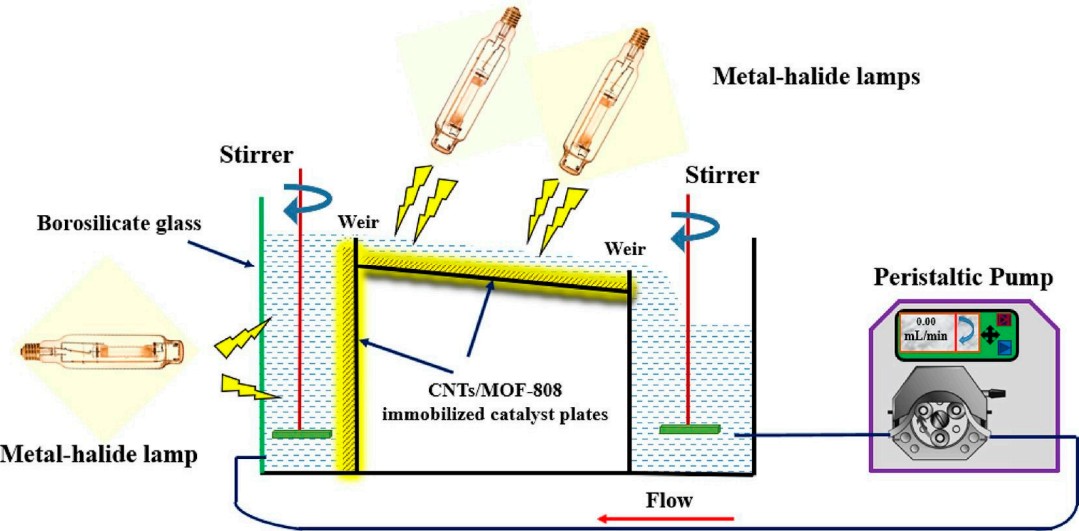

**Figure 5.** Novel photocatalytic reactor with CNTs/MOF-808 painted plates [119].

*4.3. Sustainable Approach and Systems Versatility*

Generally speaking, the immobilized system can allow high mass transfer rates when operated at high flow rates assisted by additional mechanical stirring, or by specific reactor design geometry (like sudden changes in flow direction). Such well-mixed photoreactors with immobilized catalytic layers can be regarded as plug flow reactors, a model that can be extended to an axial dispersion model for a higher range of velocities [121].

A good example of the successful implementation of both systems was recently reported by Sraw et al. (2022). Their study focused on the degradation of monocrotophos, an organophosphorus pesticide used for cotton crops [122]. Promising results were obtained with both slurry and immobilized $TiO_2$ (P25) and $W-TiO_2$ catalysts, as will be briefly presented here. Specifically, the dispersed system comprised a magnetically stirred hemispherical-shaped shallow pond slurry reactor, with a capacity of 1 L, covered with transparent polyethene film [123] (Figure 6a). The experiments were conducted at room temperature, under UV light conditions (8 blue-black UV fluorescent lamps, 20 W, mounted in parallel), and under solar irradiation. The catalysts were then immobilized on Clay beds and used in a fixed bed recirculating reactor (Figure 6b), with a working capacity of 1 L, comprising three concentric cylindrical jackets of borosil glass with one UV lamp, 20 W axially mounted within the central tube [124]. A submersible pump was used for pesticide solution recycling in the reactor. The best results were obtained for doped $TiO_2$ with W under solar irradiation: in the slurry system a mineralization degree of 96%, and a bit lower in the case of the immobilized reactor, 92%.

Higher mineralization degrees in slurry reactors are expected, mainly due to higher catalyst surface availability, but good results could be obtained in immobilized systems, as was experimentally proved. Furthermore, subsequent separation of the catalyst is not needed in the second case.

When developing novel photoreactors, the main issue is represented by the scale-up. An example of a successful scale-up started with a 2 L experimental photochemical reactor, equipped with an 8 W low-pressure mercury lamp (Figure 7a), operated in batch recirculation mode, and periodically bubbled with air [125]. Different commercial $TiO_2$ nanopowders were used as catalysts, and sodium peroxydisulfate (98%) as electron acceptors. The homogeneity of the suspension was maintained by continuous magnetic stirring and water recirculation. The system achieved in specific conditions the degradation of two worldwide used herbicides, metamitron, and metribuzin, in less than 30 min of illumination.

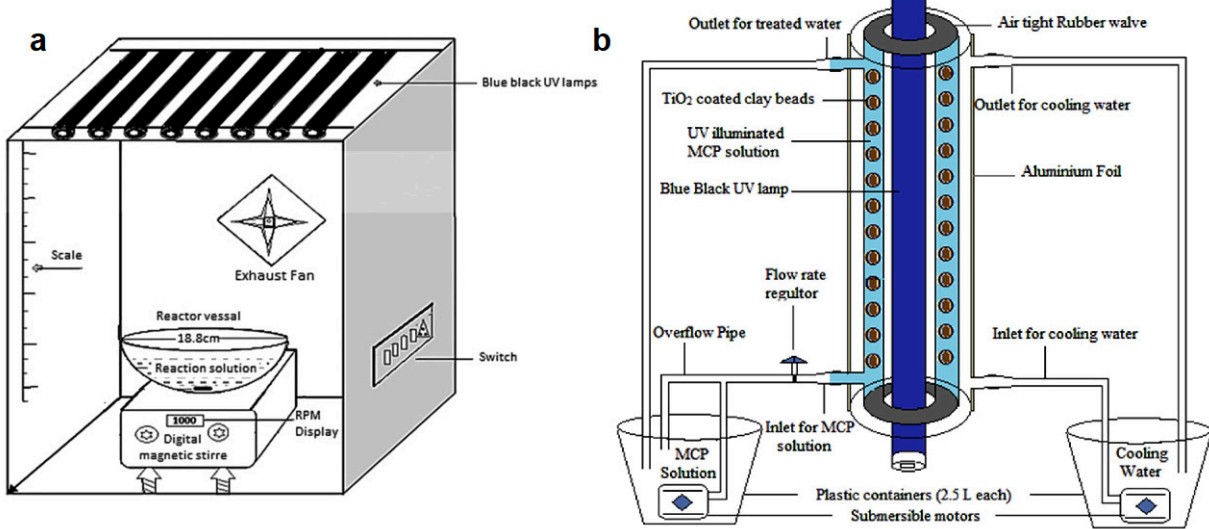

**Figure 6.** Photoreactors for pesticides degradation (**a**) Slurry reactor [123]; (**b**) Fixed bed recirculation reactor [124].

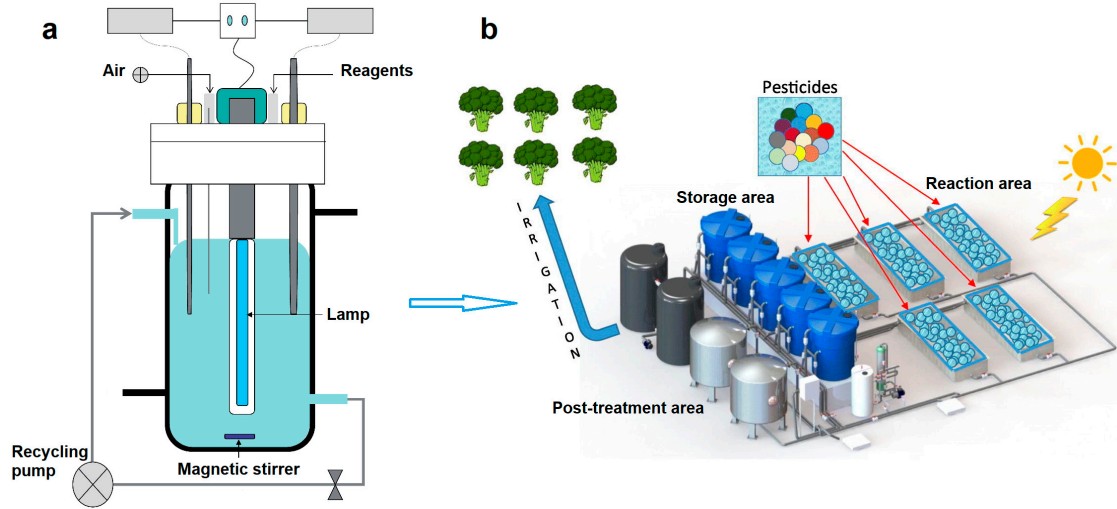

**Figure 7.** Reclamation of agro-wastewater polluted with pesticides: (**a**) Schematic drawing of the laboratory photochemical reactor [125], and (**b**) Scheme for the pilot-plant [126].

Based on the laboratory-obtained results, a pilot plant (presented in Figure 7b) was developed for the reclamation of wastewater containing a mixture of twelve or more pesticides, using natural sunlight irradiation [126,127]. The reaction unit consists of five open reaction tanks, each designed to treat up to 1000 L water, covered with high-density polyethylene, a water circulation pump, and an aeration system. $TiO_2$ nanopowder is recovered using a membrane filtration unit. The reactors, operated at ambient temperature and pressure, proved effective and low-cost degradation of persistent pesticides in acceptable time even in winter (4 days), allowing treated water to be reused for irrigation.

However, disregarding all advantages and disadvantages mentioned above for each configuration, the keys remain related to reactor integration in a complete degradation system (as a separate unit or as a hybrid equipment), and to the implementation of innovative designs as an alternative to conventional reactors. One example in this sense will be given here: the photocatalytic step reactor (PSR). Classified as a Thin Film Deposited reactor, and initially introduced as an alternative to the Thin Film Slurry reactor, the PSR offers a very large illumination area per reactor unit of volume. It is, in fact, a falling film photoreactor consisting of six steps of the same dimensions, coated with a thin layer

of the photocatalyst. As a direct consequence of a very large catalytic area, PSR allows optimal utilization of light and oxygen [128,129]. This unique configuration, as presented in Figure 8, proved efficient for both pesticide and antibiotic degradation. Thus, in an earlier study, PSR was used for the irradiation of pure pesticide solutions and diluted commercial pesticide solutions, which contained other additives. Three UV lamps Philips PL-L24W/10/4P ($\lambda$max = 365 nm) (about 38 W m$^{-2}$) were used. The obtained results indicated complete degradation of pesticides in pure solutions and high mineralization degrees: 75 % for chlortoluron and 60 % for cyproconazole after 24 h of reaction. In the case of commercial solutions, de-degradation was significantly affected by the presence of other substances/additives [130]. Another study aimed at the degradation modeling of meto-lachlor (a widely used herbicide) and chlortoluron (a widely used pesticide) degradation under a UV irradiance of about 38 W m$^{-2}$, using the same reactor configuration [128].

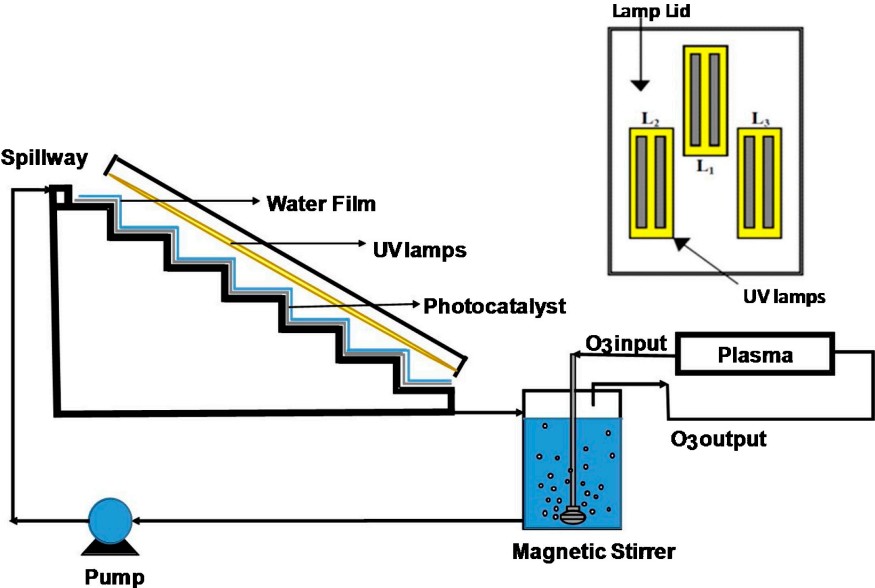

**Figure 8.** Scheme of a photocatalytic step reactor (falling film) [129].

Disregarding the approach in terms of system configuration, efficient pesticides degradation in photocatalytic processes was proved in numerous researches [131]. If proper conditions are carefully selected, the transformation of these hazardous pollutants into non-toxic compounds like $H_2O$ and $CO_2$ (complete mineralization) can be achieved in many cases.

## 5. General Conclusions and Perspectives

The constantly increasing number of researchers are focusing on improving the selectivity and yield of photocatalytic systems.

A lot of attention has been given so far to the synthesis of photocatalytic systems, including creating deliberate defects in the structure of the photocatalytic material to improve the light absorption range by promoting electrons from VB to CB or to prevent the recombination of electron-hole pairs. The great consequence of these strategies is related to the formation of more reactive species capable to degrade any organic pollutant.

In terms of slurry reactors, the great advantage remains the versatility of the configurations of these reactors that allow the degradation of any organic pollutant, while one of the major disadvantages remains the difficulty to recover the photocatalyst. The immobilized reactors offer an alternative to some issues of the slurry ones since the photocatalyst can be reused. However, until now, few photocatalytic reactor configurations were designed to treat large amounts of pesticide-contaminated water, mostly still being at a laboratory or pilot scale.

Considering all mentioned above and based on published results one general conclusion can be drawn: there is no best solution or best configuration reactor. For the degradation of a specific pollutant, any configuration can be implemented. The careful selection and optimization of each influencing parameter will determine successful degradation or complete mineralization.

Despite all efforts, industrial photochemistry is still struggling to become technically and economically sustainable. In search for these goals, innovative photoreactor designs or the development of hybrid systems should be pursued, with sequential or integrated photocatalytic units. Furthermore, in our opinion, the attention should focus on solar-based photocatalytic systems, as solar energy usage will bring economic advantages over UV light utilization.

**Author Contributions:** Conceptualization, G.O.I. and I.-M.D.; writing G.O.I., A.M. and I.-M.D. All authors have read and agreed to the published version of the manuscript.

**Funding:** This research received no external funding.

**Data Availability Statement:** Not applicable.

**Conflicts of Interest:** The authors declare no conflict of interest.

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
