# Peer review of "A Brief Review of Photocatalytic Reactors Used for Persistent Pesticides Degradation"

_2305-7084, doi:10.3390/chemengineering6060089_

Round 1

Reviewer 1 Report

The authors attempt to summarize the most recent knowledge about the use of catalysts and photocatalytic reactors for water remediation via pesticides degradation.

1- There exist many reviews of reference values about this field and other similar finding have already been reported by the other researchers. Please highlight more the novelty of this review paper.

2- The review is a little light. Please add more data and scientific descriptions on mineralization yield with some configurations

3- Please add more data on photocatalytic applications for waste water treatment.  I suggest adding the reference: Molecules. 2021; 26(6):1687. https://doi.org/10.3390/molecules26061687

4- Figure2: Please add more configurations like falling film reactor (Please read this paper: (Chemical Engineering and Processing: Process Intensification 122, 213-221 (2017))

5-Conclusions: what are the main take home messages (sustainable techniques? Weaknesses of photocatalysis for pesticides remediation ?)

6- Please add your viewpoint? solutions to remedy the weakness of each techniques of remediation in wastewater treatment. Journal of ChemEngineering_MDPI readers expect from this paper a clear vision of the direction / photocatalytic configurations of reactors & catalyst/ in the future.

Author Response

We, the Authors of the manuscript initially entitled “Photocatalytic reactors for persistent pesticides degradation” submitted to ChemEngineering Journal thank you for taking the time to analyze our work. We are deeply grateful for the received observations and comments, which significantly increased the quality of our manuscript in our opinion.

Please find below, in separate files, the answers to all comments and suggestions, in red color.

All changes were highlighted in blue inside the text of the revised manuscript. Some additional, not highlighted, minor changes (grammar, spelling) were also implemented in text, without altering the scientific content.

As a result of Manuscript revision, the Reference list was updated. Now we have a total of 131 entries. New ones are marked in blue color.

Reviewer 2 Report

This manuscript extensively summarizes the current photocatalytic methods used for the application of persistent pesticide degradation with sufficient background and explicit description in terms of degradation mechanism, photocatalyst selection, and different types of reactors. Challenges and considerations for improvements were also discussed in a scientific way based on the knowledge of authors, for guidance for researchers in the niche. I suggest the manuscript be accepted after a minor revision.

1. The keywords section includes ‘slurry systems’ and ‘immobilization’, while those are not mentioned or elucidated in the abstract part of the manuscript. Apart from that, these words are not good choices to help the audience get the main point of the review and thus, need to be changed.

2. Format (i.e., line space) of the reference section should be adjusted according to the template to accommodate the references in a more confined space.  

Author Response

(The authors gave the same response as above.)

Reviewer 3 Report

Main question addressed by the review: The work addresses the photocatalytic reactors for persistent pesticides degradation. The word review should be added to the title.

Originality and relevance of the topic: The topic is relevant to the field and it considers a potential research gap.
Added value of the paper:  The manuscript takes into account the efficiency of catalysts, mechanisms, degradation rate and configurations for optimization. However the main goal of the review is not completely justified with why those aspects are more important than other operational conditions.  

Quality of tables and tables: Good and easy to follow.
Specific improvements for the paper to be considered:

  1. Overall the review is too descriptive. First section when describing the different mechanisms, a summary would be needed and equations should be added to this discussion. 
  2. In the photocatalyst section, little discussion is considered. there are many aspects to be compared. This should be improved further. Summary table needed for clarification.
  3. Last section with configuration has a clear summary table but little critical discussion is included. The clear justification of the two optimal configurations shown afterwards with benefits should be further explained.
  4. Overall the review is not so critical and it looks like more descriptive.
  5. The conclusions are poor and they would need more elaboration so they clearly match the results.

Author Response

(The authors gave the same response as above.)

Reviewer 4 Report

Recently, I received an article for review article “Photocatalytic reactors for persistent pesticides degradation” by Gabriela Olimpia Isopencu, Alexandra Mocanu, and Iuliana-Mihaela Deleanu. I read this work with great interest. The authors consistently and systematically discussed the impact of the pesticide degradation mechanism and catalyst properties on the design of the photoreactor. However, the authors did not manage to avoid minor shortcomings in their work, which are presented below.

1.       Page 4: Please combine paragraphs 2 and 3.

2.       Page 5, line 210: Does the word “ions” have to be capitalized?

3.       Page 7: I guess Figure 1 should be number 3.

4.       Page 9, line 319: Ballari et al. is in italics.

5.       Page 12: I guess Figure 2 should be number 6.

These shortcomings do not affect the substantive value of the work. This article is suitable for publication in the ChemEngineering journal.

Author Response

(The authors gave the same response as above.)

Round 2

Reviewer 1 Report

I can recommend the Ms for publication now.

Reviewer 3 Report

Paper has significantly improved and it should be published.